# Preserving cultural heritage: Analyzing the antifungal potential of ionic liquids tested in paper restoration

Kevin Schmitz[1]*, Sebastian Wagner[1☯], Manfred Reppke[1☯], Christian Ludwig Maier[2,3], Elisabeth Windeisen-Holzhauser[1], J. Philipp Benz[1]

**1** Wood Research Munich, TUM School of Life Sciences Weihenstephan, Technical University of Munich, Freising, Germany, **2** Faculty of Chemistry and Pharmacy, Ludwig-Maximilians-University Munich, Munich-Großhadern, Germany, **3** Nitrochemie Aschau GmbH, Aschau am Inn, Germany

☯ These authors contributed equally to this work.
* schmitz@hfm.tum.de

**Data Availability Statement:** All relevant data are within the manuscript and its Supporting Information files.

## Abstract

Early industrialization and the development of cheap production processes for paper have led to an exponential accumulation of paper-based documents during the last two centuries. Archives and libraries harbor vast amounts of ancient and modern documents and have to undertake extensive endeavors to protect them from abiotic and biotic deterioration. While services for mechanical preservation such as *ex post* de-acidification of historic documents are already commercially available, the possibilities for long-term protection of paper-based documents against fungal attack (apart from temperature and humidity control) are very limited. Novel processes for mechanical enhancement of damaged cellulosic documents use Ionic Liquids (IL) as essential process components. With some of these ILs having azole-functionalities similar to well-known fungicides such as Clotrimazole, the possibility of antifungal activities of these ILs was proposed but has not yet been experimentally confirmed. We evaluated the potency of four ILs with potential application in paper restoration for suppression of fungal growth on five relevant paper-infesting molds. The results revealed a general antifungal activity of all ILs, which increased with the size of the non-polar group. Physiological experiments and ultimate elemental analysis allowed to determine the minimal inhibitory concentration of each IL as well as the residual IL concentration in process-treated paper. These results provide valuable guidelines for IL-applications in paper restoration processes with antifungal activity as an added benefit. With azoles remaining in the paper after the process, simultaneous repair and biotic protection in treated documents could be facilitated.

## Introduction

Private and public archives harbor myriads of historic paper documents as an invaluable treasure of cultural heritage. Numerous library, archive and museum communities around the

**Funding:** This research was partly funded by the Nitrochemie Aschau GmbH (https://www. nitrochemie.com/en/nitrochemie_group/home. php). CLM is an employee of this company, supplied the ionic liquids used in this study and was involved in data interpretation and manuscript proof reading. Other than this, the funders had no role in study design, data collection and analysis, decision to publish, or preparation of the manuscript. The study design was nevertheless independent of any influence by the company. There was no additional external funding received for this study.

**Competing interests:** I have read the journal's policy and the authors of this manuscript have the following competing interests: Nitrochemie Aschau GmbH holds patents (LU93386B1; EP3339508A1) for the paper reinforcement process discussed in this work. This does not alter our adherence to PLOS ONE policies on sharing data and materials. CLM still is and KL was employed by Nitrochemie Aschau GmbH as this study was conducted. All other authors have declared that no competing interests exist.

globe have committed to the task of preserving and–if necessary–restoring these documents to prevent additional deterioration.

A large share of paper wear is caused by endogenous abiotic factors, with acidic hydrolysis as a major cause [1]. The use of acidic sizing components and pulping processes, such as the bisulfite process, which were particularly popular throughout the 20th century, left paper in an acidic state and prone to slow auto-hydrolysis of glycosidic bonds [1, 2]. Furthermore, oxidization of lignin components creates further acidification and hence contributes to progression of paper brittleness [1, 3]. As a result, industrial mass de-acidification of paper-based documents has become an important business with several players involved [4, 5]. Current paper standards promoted by library, archive & museum communities, encompassing endogenous alkaline reserves and lignin contents below 1%, may have reduced the risk of acidic auto-hydrolysis, but today's paper remains exposed to a variety of other risk factors, warranting further research in preservation and restoration [1].

In parallel to endogenous risks, the control of exogenous factors is an important counter-measure in paper preservation [6]. Exogenous abiotic factors, such as humidity and temperature for example, not only affect abiotic deterioration processes, but also facilitate microbial attack [7]. In a recent survey among 57 institutional participants from 20 different countries, Sequeira *et al.* [8] have found that 79% of the participants had to deal with fungal infestations even though common preventative measures against fungal deterioration had been undertaken. According to the authors, most of these infestations were caused by unforeseen water contact (floods, leakage, fire suppression, outdoor humidity) or failure of (micro-)climate control systems, illustrating the importance of this issue beyond (sub-)tropical areas [1]. Extensive literature on deterioration-causing microorganisms, especially fungal isolates from libraries and archives, is available [6–11]. For example, *Chaetomium globosum*, *Penicillium chrysogenum* as well as other species of the respective genera were isolated from and identified as causative for dark stains in historic documents [8]. Species such as *Aspergillus versicolor* (among *Aspergillus* species) in addition to a diverse set of *Trichoderma* species were also reported to be ubiquitously found in archives, museums and libraries [10, 11]. The scientific literature also illustrates a continuous effort in preventing and controlling fungal deterioration of paper-based documents [12–17], underlining the significant threat for cellulosic documents imposed by fungi.

As identified in a survey by Sequeira *et al.* [16], available antifungal options are not yet satisfactory–especially since no high-throughput treatments for removal or prevention of fungal infestations on cellulosic documents are available to date. A novel approach, however, promises a potential for integration of antifungal components in paper documents in a combined industrial de-acidification and mechanical paper reinforcement process [5]. In this process, regenerated cellulose fibers are solubilized in an ionic liquid (IL) and dimethyl sulfoxide (DMSO) as a co-solvent, before applying the solution to cellulosic documents in an immersion bath (Fig 1). Upon solvent exchange with hexamethyldisiloxane (HMDSO), the cellulose fibers precipitate at the cellulosic surfaces [18, 19], thereby conferring mechanical reinforcement with neglectable impairment of readability [5]. Notably, residues of (co-)solvents and the IL used for cellulose solubilization get incorporated in the paper material along the way.

ILs, also commonly referred to as liquid salts [20], consist of an organic cation or anion with an asymmetrical charge distribution (which is usually aprotic and stably charged in its nature) and a matching organic or inorganic counter ion [21]. ILs are mostly recognized as high potential solvents in green chemistry applications due to their non-volatile characteristics [22]. Interestingly, the class of ionic liquid used in the paper reinforcement process (represented by 1-butyl-3-methylimidazolium-acetate; BA) shares key structural characteristics (Fig 2) with well-known antifungal agents such as Clotrimazole [23] or Miconazole [24].

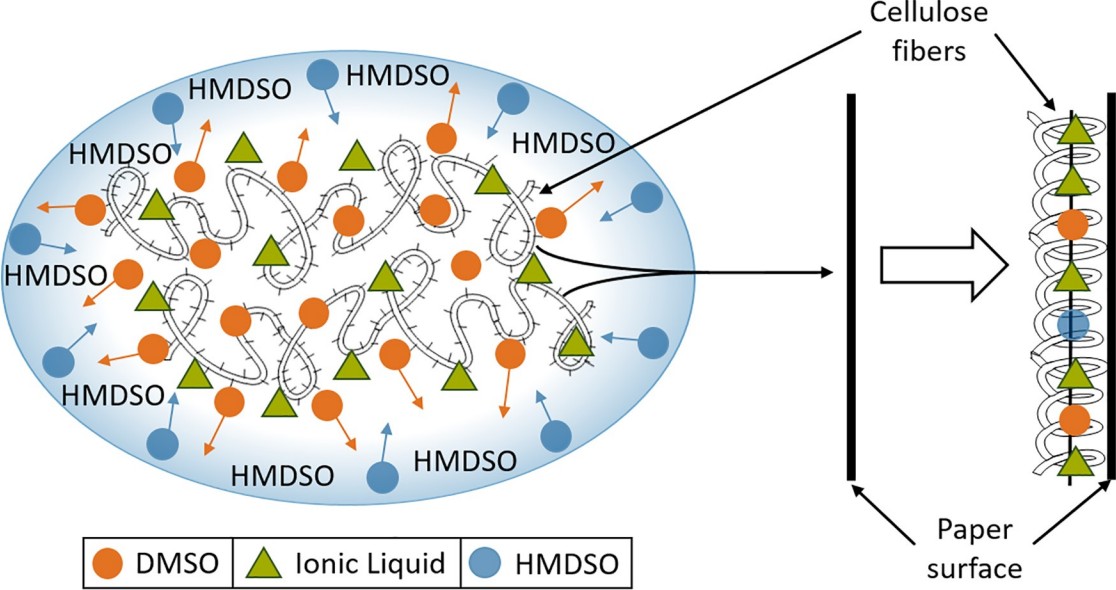

**Fig 1. Process scheme of the paper reinforcement method.** Process as described by Krupp *et al.* [18] and based on Maier [5]. Cellulose fibers are solubilized in IL (green triangles) and DMSO (orange circles) as co-solvent (left). Upon solvent exchange with HMDSO (blue circles; arrows indicate solvent replacement), cellulose fibers precipitate at cellulosic surfaces, thereby adding mechanical strength to the cellulosic documents while incorporating IL molecules (and other co-solvents).

Similarities comprise a charged, polar imidazolium group, which is known to be involved in blocking of an enzyme in ergosterol biosynthesis [25, 26], as well as a nonpolar (derivatized) aliphatic molecule tail, believed to interact with fungal membranes and thereby disturbing their integrity at higher concentrations [27]. Extrapolating these findings, ILs of this class could potentially confer antifungal properties to documents treated in the aforementioned paper reinforcement process upon incorporation into the cellulosic documents. In this study, we therefore tested four ILs matching these characteristics and showing promising process compatibility properties for their antifungal potential. We could show in bioassays that BA and HC (1-hexyl-3-methylimidazolium-chloride; Fig 2) abolished growth of all tested fungi at 10% and 1% volume concentrations, respectively. Furthermore, fungal growth was significantly impaired at lower concentrations of 1% and 0.1%, respectively. While antifungal properties could not be confirmed on BA-treated paper samples, this study nevertheless provides important guidelines for future IL selection for incorporation of fungal protection in paper restoration.

## Materials and methods

### Growth media and ionic liquids

Malt extract media (MEM) agar was prepared as 3% malt extract (Roth) and 0.5% peptone from casein (Sigma) in 1.5% agar-agar (Sigma). Minimal medium (MM) agar for fungal spore generation was prepared as 1x Vogel's salts [28] and 2% sucrose in 1.5% agar-agar (Sigma). MM agar with 2% carboxymethyl-cellulose (CMC low viscosity, Sigma Aldrich) instead of sucrose was used for ionic liquid dependent growth experiments for all fungal strains and was supplemented with varying amounts of different ILs provided by Nitrochemie Aschau GmbH (1-butyl-3-methylimidazolium acetate, 1-butyl-3-methylimidazolium chloride, 1-allyl-3-methylimidazolium-chloride, 1-hexyl-3-methylimidazolium-chloride) for some experiments

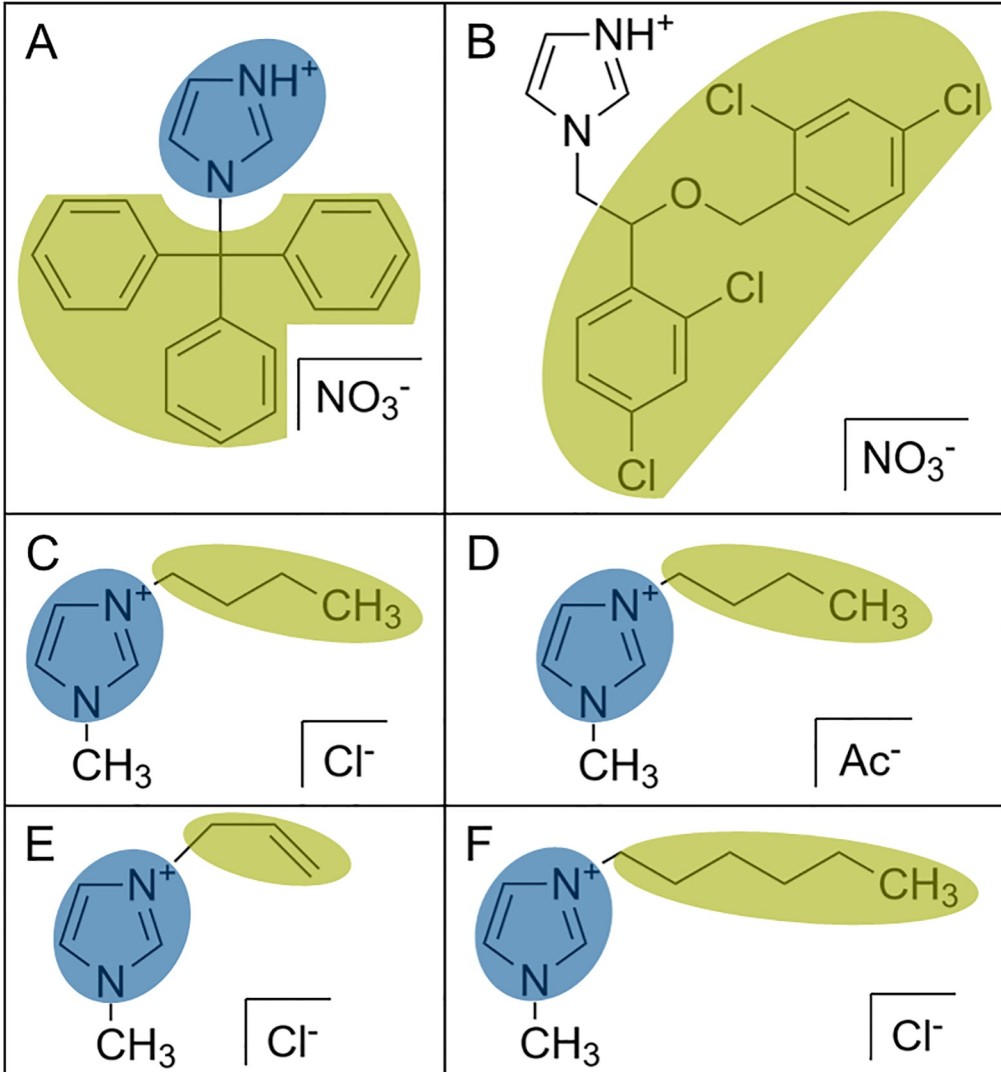

**Fig 2. Chemical structures of known antifungals and ILs.** Clotrimazole (a) and Miconazole (b) as known antifungals with polar protic imidazolium groups (blue) and a rather nonpolar, (derivatized) aliphatic molecule region (green) causing overall asymmetrical molecule properties. Ionic liquids tested for antifungal activity in this study were: 1-butyl-3-methylimidazolium with (c) chloride (BC) and (d) acetate (BA) as counter ions as well as (e) 1-allyl-3-methylimidazolium-chloride (AC) and (f) 1-hexyl-3-methylimidazolium-chloride (HC). All four ILs contain an aprotic imidazolium group (blue) as well as nonpolar tails (green) varying in size and hydrophobicity.

as stated in the main text. Paper samples treated in the aforementioned paper reinforcement process [19] with 1-butyl-3-methylimidazolium chloride or 1-butyl-3-methylimidazolium acetate as cellulose solving ionic liquids were provided by Nitrochemie Aschau GmbH. 1-Butyl-3-Methylimidazolium chloride was applied as a solution in DMSO as it was solid at room temperature.

## Fungal strains

Cultures of *Aspergillus versicolor* (DSM-1943), *Chaetomium globosum* (DSM-62109), *Penicillium chrysogenum* (DSM-244) and *Penicillium glabrum* (DSM-2017) were retrieved from the Leibnitz Institute German Collection of Microorganisms and Cell Cultures (DMSZ) and

verified via ITS sequencing of PCR products generated using primer pair ITS1 and ITS4 [29] on genomic DNA isolated via microwave treatment [30]. Retrieved ITS sequences were blasted against the fungal genome database using the National Center for Biotechnology Information (NCBI) nucleotide BLAST [31]. Perithecia formation of *C. globosum* indicated the presence of both mating types in the culture. Isolate *Trichoderma capillare* P9_B2_1 (NCBI accession number MF964230) was retrieved from an isolated stock [32].

## Harvest of fungal spores

Spores were harvested after incubation on MEM agar at room temperature (RT) under light for seven days for *Aspergillus versicolor* and *T. capillare*, while *Penicillium* species were grown on MM agar under identical conditions for spore generation. Spores were scratched off of the mycelial biomass using a toothpick, suspended in physiological sodium chloride spore solution containing 0.05% Tween® 80 and subsequently filtered through sterile Miracloth® to eliminate hyphal remainders. *Chaetomium globosum* was incubated as a culture of mixed mating types on MEM agar at RT under light for five weeks until black perithecia were formed [33]. Perithecia were selectively harvested into a 1.5 ml Eppendorf tube and vortexed with 1 ml of spore solution to release spores before the suspension was filtered accordingly. Correlation curves for spectroscopic microplate reader (Tecan® Infinite M200 Pro) optical density values at 600 nm wavelength ($OD_{600}$) and spore counts on a Neubauer counting chamber were determined for each spore type for quantification of spore inoculum volumes in successive experiments.

## Agar-based assays for antifungal IL activity and potency determination

Fungal spores were densely plated onto MM-CMC agar plates at a density of $10^5$ spores per $cm^2$. Wells were punched into the inoculated agar to locally apply 20 μl of each IL. Plates were incubated at RT under light for germination and evaluated for halo formation and stability after four, seven, 14 and 28 days. Clotrimazole (in DMSO) was applied at 10,000 fold lower concentration as a positive, DMSO as a negative control. For potency evaluation, equimolar amounts of 140 μmol of each IL as well as an aqueous dilution series thereof (75%, 50%, 25% and 10%) were spotted onto inoculated plates prepared as above in duplicates. Matching molar amounts of sodium chloride and sodium acetate were also tested for fungal growth impairment in this assay to exclude specific effects of the IL counter ions. Halo radii were measured fourfold after 14 days and plotted against the applied amount of IL. For microtiter scale experiments, the outermost wells of the plate were filled with 200 μl sterile water to provide a humidity reservoir. MM-CMC was supplemented with varying volume per volume percentages of each IL (10%, 1%, 0.1%, 0.01%, 0%), respectively. Three wells per IL and dilution step were prepared with 200 μl medium per plate. Upon Inoculation of each well with $10^4$ spores, plates were covered with a standard microtiter plate coverlid with or without aeration spacers, incubated at room temperature under light and evaluated for growth after four, seven and fourteen days.

## Calculation of IL content in paper samples via elemental analysis of nitrogen (N)

Determination of the IL-content of paper treated in the reinforcement process was carried out via elemental analysis aiming at total nitrogen quantification after complete combustion. The analyses were performed on a CHN-O Rapid Elemental Analyser (Heraeus). An increase of 0.86 basis points in nitrogen content of treated compared to untreated paper was determined. Neither of the co-solvents DMSO or HMDSO contains nitrogen, and the overall nitrogen

content of viscose fibers (Kelheim fibers GmbH) applied in the paper reinforcement process was assumed to be in a comparable (if not even lower) range compared with the original cellulosic document. Considering the nitrogen mass content of BA being 14.14%, this results in an estimated IL content of 6.1% in treated paper.

## Growth assays on paper

Fungal spores were prepared as a spore suspension at $10^6$ spores per ml in spore solution. 1 ml of this suspension was used to inoculate autoclaved paper samples of 2 cm x 2 cm in size. Inoculated paper samples were placed in individual sterile petri dishes and incubated at room temperature in the light under high humidity conditions to avoid paper desiccation. Growth was checked after four, seven ant fourteen days using a stereomicroscope.

# Results

For our study, we selected four fungal strains known to cause paper deterioration in archives, namely *Chaetomium globosum*, *Penicillium chrysogenum*, *Penicillium glabrum* and *Aspergillus versicolor* [8, 10, 11] plus one recently isolated wild type strain of the highly cellulolytic genus *Trichoderma*, *T. capillare* [32]. All of the five selected fungal strains were reported to have good cellulolytic capabilities [34–37], making them suitable for the evaluation of the antifungal potentials of ILs on paper-decomposing molds.

## General antifungal activity and comparison of antifungal potency of ILs

Principle susceptibility of all five fungal strains towards all four ILs was tested in an initial screening. To this end, all ILs were spotted onto Minimal Medium plates with carboxymethyl-cellulose as carbon source (MM-CMC) densely inoculated with fungal spores. As a representative example of all fungi, Fig 3a and 3b show susceptibility of *P. glabrum* to HC and BA, while BC and AC caused minimal to no observable effects. Stereomicroscopic analysis of the hyphal-free halos formed around HC and BA application sites revealed that germination was suppressed in these areas. Spores were not observed to germinate even after 28 days of incubation, thereby demonstrating persistence and durability of the antifungal effect.

Quantification of fungal growth impairment by equimolar amounts of each of the four ILs was conducted for comparison of the four ILs via measurement of growth-free halo sizes on inoculated agar plates. Representative results are displayed for *P. glabrum* in Fig 3c and 3d. HC clearly had the strongest effect on all fungi tested, followed by BA, while no significant effect was observed for AC and BC in accordance with previous results. Notably, *C. globosum* and *T. capillare* revealed even stronger susceptibility to HC and BA than *P. glabrum*, which levelled equal to *A. versicolor* and *P. chrysogenum* in terms of susceptibility to the tested ILs (S1 Fig). Halos were observed to be stable and not penetrated by adjacent hyphae throughout 28 days of incubation. Halo formation was not observed on plates inoculated with corresponding molar amounts of sodium chloride or sodium acetate, indicating that the different counter ions (or the ionic strength, respectively) of the tested ILs did not facilitate growth impairment or inhibition *per se*.

As an important measure with regard to the paper reinforcement process, antifungal activities of the tested ILs would best be assessed as minimal inhibitory concentrations (MIC) in volume percentages. For this, a miniaturized growth assay was developed that allowed assessment of MIC and minimal growth impairing concentration (MGIC). MM-CMC with varying IL concentrations was inoculated with fungal spores in microtiter plate wells (Fig 3e and 3f). All ILs but BC caused clear inhibition of growth at 10% (v/v) concentration, while BC could only mildly impair but not abolish growth at 10% in *P. chrysogenum* and *P. glabrum*. Apart from

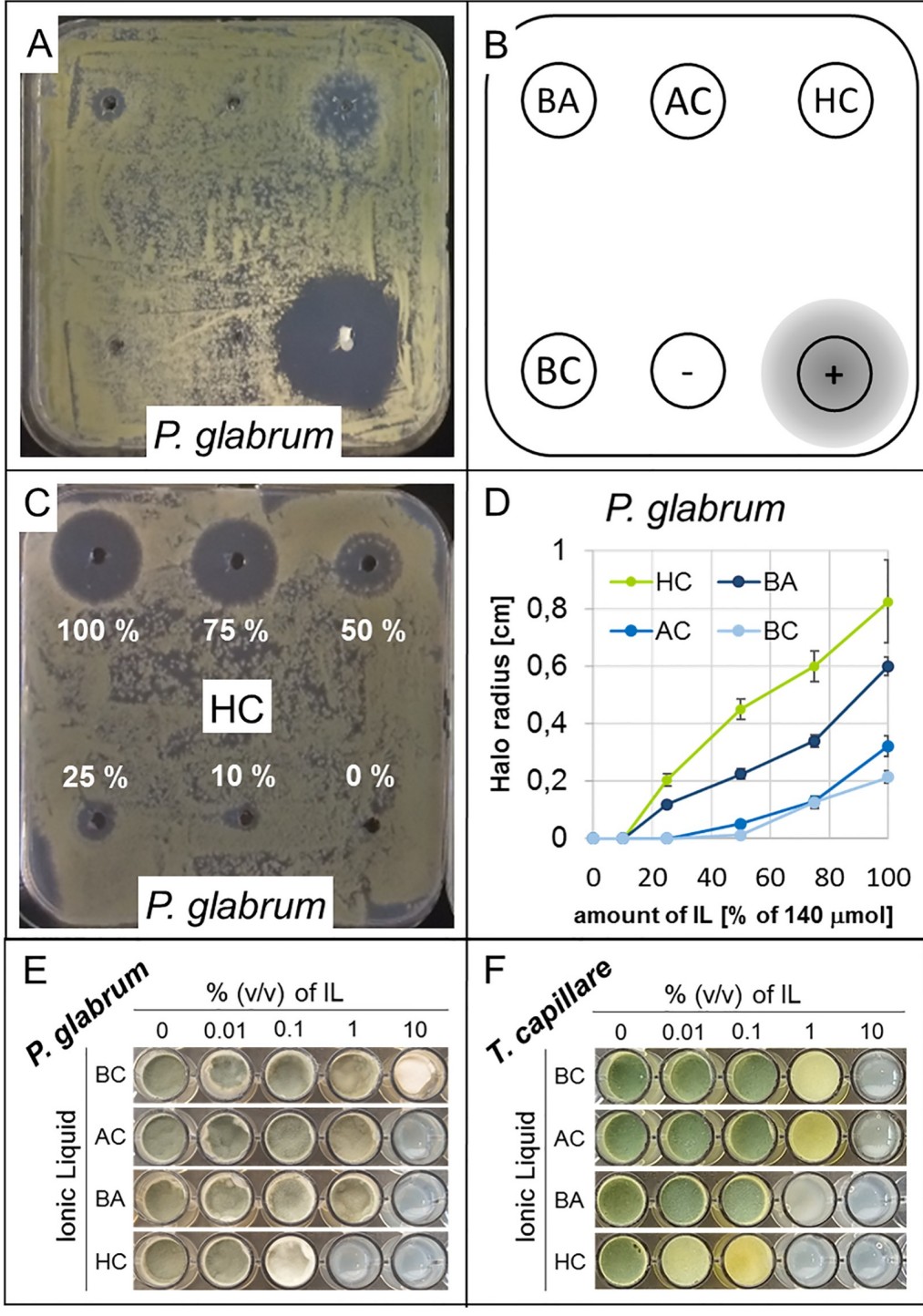

**Fig 3. Principal investigation and comparison of antifungal IL activity.** (a) shows the exemplary result of the initial antifungal activity screening of all tested ILs including DMSO as a negative and Clotrimazole as a positive control on densely inoculated MM-CMC plates for *P. glabrum*, spotted following scheme (b), after seven days of incubation. c) and d) demonstrate the molar potency comparisons of the four ILs, using *P. glabrum* spot dilution series of HC (c) and the halo radius plot for all four ILs (d) as representative data for all tested fungi. e) MIC and MGIC concentration analysis in the microtiter scale growth assay for *P. glabrum* and for the most susceptible fungal strain in this study, *T. capillare* (f).

HC, which caused strong growth impairment and reduced sporulation already at the 1% concentration level in all tested fungi (and even completely inhibited germination of *T. capillare*, *P. glabrum* and *C. globosum*), only BA was observed to show mild growth and sporulation impairment at the 1% concentration level in some of the tested fungi, especially in *T. capillare* and *C. globosum* (S2 Fig). In summary, BA and especially HC were determined to have the highest antifungal potential.

### Testing fungal growth on IL-treated *vs.* non-treated paper as carbon source

Elemental analysis revealed an IL content of about 6.1% (w/w) in treated paper (see Methods). With an observed MIC value of around 1% for HC and MGIC of 1% for BA, antifungal effects should therefore, in principle, be observable in incubation experiments of fungal spores on BA- and HC-treated paper samples. Since AC and HC had shown poor cellulose solubilizing properties, the paper reinforcement process could only be established with BC and BA in pilot scale so far. Hence, no AC- or HC-treated paper could be tested for antifungal activity in this study.

When BA-treated and spore-inoculated paper samples were incubated on water agar plates, no qualitative or quantitative effect in growth impairment was observed for *T. capillare*, which had been found to have the strongest susceptibility to all tested ILs in previous experiments (Fig 4a). To obviate a possible dilution effect of the effective IL concentration through passive diffusion from the treated paper into the water agar, we also tested whether growth impairment could be observed during incubation of inoculated paper samples without agar but under high humidity conditions. Again, no perceptible difference in growth of *T. capillare* was observed on untreated versus BA-treated paper samples (Fig 4b), imposing a need for additional investigation in subsequent studies.

## Discussion

We were able to demonstrate that all tested ILs exhibited growth-inhibiting effects on relevant paper-deteriorating molds at high concentrations of 10% in an agar-based assay, while BC

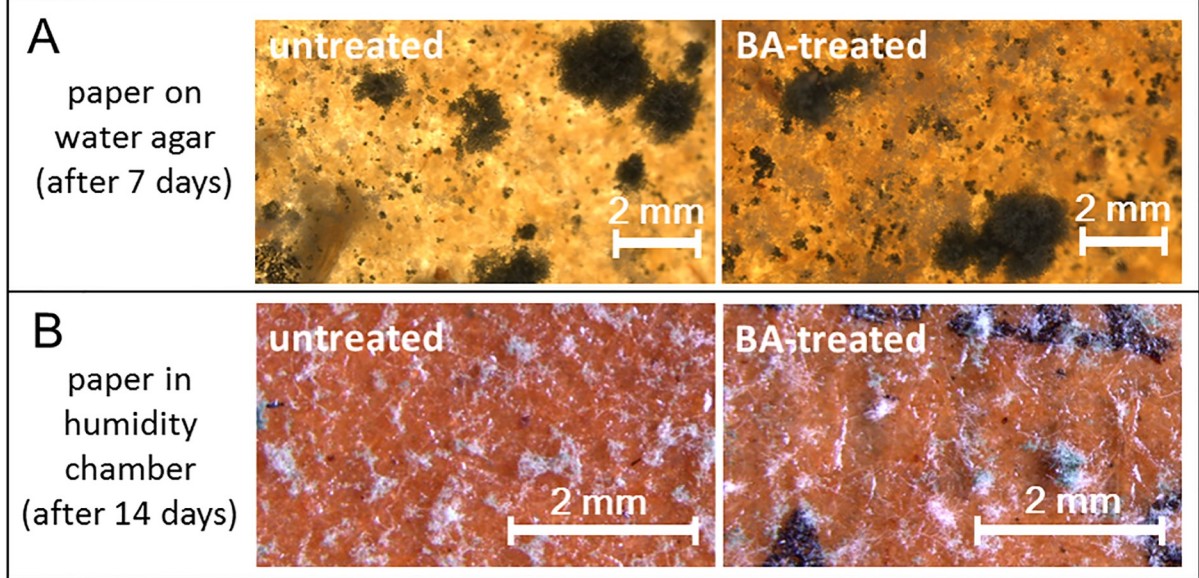

**Fig 4. Fungal growth tests on IL-treated paper samples.** *T. capillare* spores inoculated on BA-treated (right) or untreated (left) paper and incubated on water agar (a) or under saturated humidity conditions in a high humidity chamber (b) did not show growth impairment or inhibition effects after paper treatment.

could only impair growth of the two *Penicillia* under these conditions. Potency experiments considering halo sizes on IL-inoculated agar plates and determining minimal inhibiting concentrations (MIC) as well as minimal growth impairment concentrations (MGIC) further revealed HC to be the strongest antifungal substance of the tested set, followed by BA, AC and BC. Detailed evaluation of the IL effects revealed that the order of their respective potency matches well with the size of their corresponding non-polar group, which is pointing towards the importance of the hydrophobic region for antifungal activity. Similar results were observed when the membrane-disturbing effects of three antifungal imidazolium drugs were tested [27], indicating that larger nonpolar groups may facilitate stronger membrane interactions and hence better antifungal activities in imidazolium-containing ILs. A systematic study focusing purely on the dependency of the antifungal effect on non-polar group sizes and structures would likely help to elucidate this correlation further.

Our study also allowed for hypothesis generation on antifungal effects of IL characteristics beyond the non-polar group constitution. Remarkably, the two ILs BA and BC only differ in their respective counter ions acetate and chloride, which by themselves were shown not to impair or inhibit fungal growth at equivalent concentrations in this study. It remains unclear, however, why the antifungal activity of BA was quite prominent while that of BC was very poor. Literature on plant cuticular membrane penetration of glyphosate salts for example indicates that the nature of the counter ion indeed influences the membrane penetration characteristics [38]. Fungal cell membrane penetration would be a crucial prerequisite for inhibition of ergosterol biosynthesis as a proposed mode of antifungal action of imidazolium ILs similar to that described for established imidazolium antifungals such as Clotrimazole [25, 26]. Additional research should address the influence of different counter ions in imidazolium ILs on antifungal activity, which could further benefit the development of highly potent antifungal drugs.

Considering an application as an antifungal component in paper restoration, the persistence of growth-free halos around IL inoculation sites throughout the observation period of four weeks indicated long-term stability and inertness of the tested ILs towards fungal decay or abiotic factors, which would be a beneficial trait for archived documents. However, while strong growth impairment was observed for *T. capillare* on MM-CMC containing 1% (v/v) of BA, no such growth impairment could be observed on BA-treated paper–which was found to contain about 6% (w/w) of BA in elemental analysis. Moreover, replacing the agar plates with a high-humidity chamber, as a precaution to avoid the potential dilution of the effective IL concentration from the paper samples, did not result in perceptible growth differences. Although the actual cause for the absence of growth impairment on BA-treated paper could not finally be assessed in this study, we hypothesize that the IL might be trapped inside the adsorbed cellulose fibers, rendering it poorly available for the fungus. Effective IL amounts released during fungal enzymatic cellulose degradation might thus be too low to cause notable growth impairment. However, further studies need to be conducted to examine this hypothesis.

Apart from IL characteristics like non-polar group size and the type of counter ion identified in this study as potential optimization targets for further studies on antifungal IL activity, we also recommend the consideration of adapted strategies to implement antifungals in mechanical paper restoration. For example, the antifungal drug Clotrimazole–which is structurally highly related to the tested ILs–displayed more than 1000 times higher activity compared to HC, the most potent IL tested in this study. However, Clotrimazole and related known antifungal compounds alone are not suitable for substitution of the cellulose-solubilizing ILs in the industrial process. Additional application designs should therefore also evaluate the possibility of applying highly potent antifungals as additional additives in the established

organic solvent-based mechanical restoration processes to deliver them to the cellulosic material alongside the fiber repair processes.

Altogether, by demonstrating a general antifungal potential of ILs and identifying initial chemical parameters that appear to affect their activity, this study can serve as a guidance to focus ongoing research on the integration of biotic preservation substances into mechanical restoration processes for cellulosic documents–and hence contribute to biotic preservation of cultural heritage.

## Ethics and consent

This article does not contain any studies with human participants or animals performed by any of the authors.

## Supporting information

**S1 Fig. Molar potency evaluation of ILs on remaining fungi via halo size analysis.** As seen for *P. glabrum*, HC had the strongest effect on all tested fungi, followed by BA, AC and BC. *T. capillare* (a) did show the strongest susceptibility to the tested ILs, followed by *C. globosum* (b), *P. glabrum*, *A. versicolor* (c) and *P. chrysogenum* (d).
(TIF)

**S2 Fig. MIC and MGIC concentration analysis in the microtiter scale growth assay.** HC inhibited growth of *C. globosum* at 1% (v/v) and impaired growth of *A. versicolor* and *P. chrysogenum* at this concentration. BA was further shown to mildly impair growth of C. globosum at 1% (v/v). All ILs but BC were able to inhibit growth of all tested fungi at 10% concentration, stringently.
(TIF)

## Acknowledgments

We want to acknowledge Dr. Klaus Langerbeins for strategic research and funding support. We furthermore want to thank Petra Arnold, Sabrina Paulus, Nadine Griesbacher and Claudia Strobel for excellent technical assistance.

## Author Contributions

**Conceptualization:** Kevin Schmitz, Christian Ludwig Maier, J. Philipp Benz.

**Formal analysis:** Sebastian Wagner.

**Funding acquisition:** J. Philipp Benz.

**Investigation:** Kevin Schmitz, Sebastian Wagner, Manfred Reppke, Elisabeth Windeisen-Holzhauser.

**Methodology:** Kevin Schmitz, Sebastian Wagner, Manfred Reppke, Elisabeth Windeisen-Holzhauser, J. Philipp Benz.

**Project administration:** J. Philipp Benz.

**Resources:** Christian Ludwig Maier, Elisabeth Windeisen-Holzhauser, J. Philipp Benz.

**Supervision:** Kevin Schmitz, J. Philipp Benz.

**Validation:** Kevin Schmitz, Manfred Reppke, Elisabeth Windeisen-Holzhauser.

**Visualization:** Kevin Schmitz, Sebastian Wagner, Manfred Reppke, Christian Ludwig Maier.

**Writing – original draft:** Kevin Schmitz.

**Writing – review & editing:** Kevin Schmitz, Sebastian Wagner, Manfred Reppke, Christian Ludwig Maier, Elisabeth Windeisen-Holzhauser, J. Philipp Benz.

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
