## [Decision Letter · Decision Letter 0]

7 Aug 2019

PONE-D-19-17935

Preserving cultural heritage: Analyzing the antifungal potential of ionic liquids tested in paper restoration

PLOS ONE

Dear M.Sc. Schmitz,

Thank you for submitting your manuscript to PLOS ONE. After careful consideration, we feel that it has merit but does not fully meet PLOS ONE’s publication criteria as it currently stands. Therefore, we invite you to submit a revised version of the manuscript that addresses the points raised during the review process.

We would appreciate receiving your revised manuscript by Sep 21 2019 11:59PM. To enhance the reproducibility of your results, we recommend that if applicable you deposit your laboratory protocols in protocols.io, where a protocol can be assigned its own identifier (DOI) such that it can be cited independently in the future. For instructions see: http://journals.plos.org/plosone/s/submission-guidelines#loc-laboratory-protocols

We look forward to receiving your revised manuscript.

Kind regards,

Binod Bihari Sahu, Ph.D.

Academic Editor

PLOS ONE

Journal Requirements:

This research was partly funded by the Nitrochemie Aschau GmbH (https://www.nitrochemie.com/en/nitrochemie_group/home.php).

Christian L. Maier is an employee of this company, supplied the ionic liquids used in this study and was involved in data interpretation and manuscript proof reading. The study design was nevertheless independent of any influence by the company.

We note that you received funding from a commercial source:Nitrochemie Aschau GmbH

3. We note that you have a patent relating to material pertinent to this article. Please provide an amended statement of Competing Interests to declare this patent (with details including name and number), along with any other relevant declarations relating to employment, consultancy, patents, products in development or modified products etc. Please confirm that this does not alter your adherence to all PLOS ONE policies on sharing data and materials, as detailed online in our guide for authors http://journals.plos.org/plosone/s/competing-interests by including the following statement: "This does not alter our adherence to  PLOS ONE policies on sharing data and materials.” If there are restrictions on sharing of data and/or materials, please state these. Please note that we cannot proceed with consideration of your article until this information has been declared.Please know it is PLOS ONE policy for corresponding authors to declare, on behalf of all authors, all potential competing interests for the purposes of transparency. PLOS defines a competing interest as anything that interferes with, or could reasonably be perceived as interfering with, the full and objective presentation, peer review, editorial decision-making, or publication of research or non-research articles submitted to one of the journals. Competing interests can be financial or non-financial, professional, or personal. Competing interests can arise in relationship to an organization or another person. Please follow this link to our website for more details on competing interests: http://journals.plos.org/plosone/s/competing-interests

Reviewers' comments:

Reviewer's Responses to Questions

**Comments to the Author**

1. Is the manuscript technically sound, and do the data support the conclusions?

Reviewer #1: Yes

Reviewer #2: Yes

2. Has the statistical analysis been performed appropriately and rigorously? 

Reviewer #1: Yes

Reviewer #2: I Don't Know

3. Have the authors made all data underlying the findings in their manuscript fully available?

Reviewer #1: Yes

Reviewer #2: Yes

4. Is the manuscript presented in an intelligible fashion and written in standard English?

Reviewer #1: Yes

Reviewer #2: Yes

5. Review Comments to the Author

Reviewer #1: The manuscript presented by Schmitz entitled " Preserving cultural heritage: Analyzing the antifungal

potential of ionic liquids tested in paper restoration " can be accepted after clarification of the following points:

1. The experiments are performed taking DMSO as the medium which itself shows zwiterionic properties and bear a appreciable value of polarity. So inorder to understand the effects of the planted Ionic liquids, other solvent mediums like ethanol, acetone, ethers may be done in presence of ionic liquids. Also blank tests in these solvents have to be done and reported with comparisons with DMSO.

2. The authors have presented a model of the spatial orientation of the ionic liquid, solvent on cellular fibres on paper in Fig.2. The logical design may be explained clearly.

Reviewer #2: Paper is good and very much applied aspects. It definitely, open a new area of research in the field of conservation work. But, personally, I feel if there is scope for modification, specially in discussion section.

6. PLOS authors have the option to publish the peer review history of their article (what does this mean?). If published, this will include your full peer review and any attached files.

Reviewer #1: Yes: Debayan Sarkar

Reviewer #2: Yes: Ramesh Sahani

---

## [Author Response · Author response to Decision Letter 0]

21 Aug 2019

To the editor:

We have made the amended formal changes and included required statements in the Funding and Competing Interest sections as requested (Editor Requests 1-3).

------------------

To the reviewers:

We would like to gratefully thank you for your kind appreciation of our work and for your critical review of our article "Preserving cultural heritage: Analyzing the antifungal potential of ionic liquids tested in paper restoration". We value your constructive remarks and would like to respond to them point-by-point in the following:

To reviewer #1:

"The manuscript presented by Schmitz entitled "Preserving cultural heritage: Analyzing the antifungal potential of ionic liquids tested in paper restoration" can be accepted after clarification of the following points:

1. The experiments are performed taking DMSO as the medium which itself shows zwitterionic properties and bear a appreciable value of polarity. So in order to understand the effects of the planted Ionic liquids, other solvent mediums like ethanol, acetone, ethers may be done in presence of ionic liquids. Also blank tests in these solvents have to be done and reported with comparisons with DMSO."

Answer: We highly appreciate your constructive suggestions. However, as the original process, in which our ILs are applied, involves initial solubilization and subsequent precipitation of cellulose onto printed document surfaces, we are tightly limited in terms of the solvents we can use. Ethanol and acetone, for example, do not facilitate solubilization of cellulose fibers and strongly impair legibility of the treated documents due solubilization of the ink. We would also like to clarify that DMSO was only used as a solvent for Clotrimazole and 1-Butyl-3-Methylimidazolium chloride (both solid at room temperature), while the other ionic liquids were applied as pure liquids (without the need for additional solvents during potency assessment in this study). In order to clarify the methodological sequence for the reader, we made some amendments in the methods section. Moreover, a blank test of the antifungal activity of DMSO had already been included in the initial antifungal activity screening (Figure 3 A, B). Overall, we therefore do not think that additional tests with further solvents would give relevant results that could actually be used in the process.

"2. The authors have presented a model of the spatial orientation of the ionic liquid, solvent on cellular fibers on paper in Fig.2. The logical design may be explained clearly."

Answer: We believe that the reviewer is actually referring to Figure 1 and – after reconsideration - agree with the reviewer that the initial illustration was partially misleading in terms of how the ionic liquid and DMSO entities were positioned. We have therefore now adapted the figure including its figure legend accordingly. We believe that the figure is now more accurately representing the process.

To reviewer #2: 

Paper is good and very much applied aspects. It definitely open a new area of research in the field of conservation work. But, personally, I feel if there is scope for modification, especially in discussion section."

Answer: We have gladly modified the discussion section and hope to have improved it by doing so. Firstly, we have re-arranged the discussion paragraphs to better mirror the presentation sequence of the results. We believe that this allows a better orientation for the reader and hence could benefit the level of comprehensiveness. 

We have rearranged the last bit of the first paragraph (i) to bring the sections discussing the correlation of IL potency and IL structure closer together in the text and (ii) to be able to discuss the long-term stability of the antifungal activity in the broader context of the paper-based application.

We furthermore included a statement in the second paragraph that points out the necessity for further studies on the contribution of non-polar group size and structure to the respective antifungal activity of an ionic liquid – as was done in the third paragraph as well. 

Lastly, we extended the second-to-last paragraph. We aimed at putting stronger emphasis on the proposed future directions for research on biotic preservation implementation in paper restoration processes. 

Additional minor modifications were made to improve overall readability and to clarify certain statements.

---

## [Decision Letter · Decision Letter 1]

27 Aug 2019

[EXSCINDED]

Preserving cultural heritage: Analyzing the antifungal potential of ionic liquids tested in paper restoration

PONE-D-19-17935R1

Dear Dr. Schmitz,

We are pleased to inform you that your manuscript has been judged scientifically suitable for publication and will be formally accepted for publication once it complies with all outstanding technical requirements.

With kind regards,

Binod Bihari Sahu, Ph.D.

Academic Editor

PLOS ONE

Additional Editor Comments (optional):

Reviewers' comments:

Reviewer's Responses to Questions

**Comments to the Author**

1. If the authors have adequately addressed your comments raised in a previous round of review and you feel that this manuscript is now acceptable for publication, you may indicate that here to bypass the “Comments to the Author” section, enter your conflict of interest statement in the “Confidential to Editor” section, and submit your "Accept" recommendation.

Reviewer #1: All comments have been addressed

2. Is the manuscript technically sound, and do the data support the conclusions?

Reviewer #1: Yes

3. Has the statistical analysis been performed appropriately and rigorously? 

Reviewer #1: Yes

4. Have the authors made all data underlying the findings in their manuscript fully available?

Reviewer #1: Yes

5. Is the manuscript presented in an intelligible fashion and written in standard English?

Reviewer #1: Yes

6. Review Comments to the Author

Reviewer #1: The suggested corrections with details have been incorporated. The paper can be accepted for publication.

7. PLOS authors have the option to publish the peer review history of their article (what does this mean?). If published, this will include your full peer review and any attached files.

Reviewer #1: No

---

## [Editor Report · Acceptance letter]

9 Sep 2019

PONE-D-19-17935R1 

Preserving cultural heritage: Analyzing the antifungal potential of ionic liquids tested in paper restoration 

Dear Dr. Schmitz:

I am pleased to inform you that your manuscript has been deemed suitable for publication in PLOS ONE. Congratulations! Your manuscript is now with our production department. 

With kind regards,

on behalf of

Dr. Binod Bihari Sahu 

Academic Editor

PLOS ONE